# Mechanical Performance of 3D-Printed Polyethylene Fibers and Their Durability against Degradation

**DOI:** 10.3390/ma16145182

**Published:** 2023-07-24

**Authors:** Yao Xiao, Shikai Zhang, Jingyi Chen, Baoling Guo, Dong Chen

**Affiliations:** 1Department of Oncology, Longyan First Affiliated Hospital of Fujian Medical University, Longyan 364000, China; 11927061@zju.edu.cn; 2State Key Laboratory of Clean Energy Utilization, College of Energy Engineering, Zhejiang University, Hangzhou 310003, China; skzhang@zju.edu.cn (S.Z.); jchen1@g.harvard.edu (J.C.); 3John A. Paulson School of Engineering and Applied Sciences, Harvard University, Cambridge, MA 02138, USA; 4Zhejiang Key Laboratory of Smart Biomaterials, College of Chemical and Biological Engineering, Zhejiang University, Hangzhou 310027, China

**Keywords:** polyethylene, FDM-printed PE fibers, mechanical properties, degradation

## Abstract

Polyethylene (PE), one of the most popular thermoplastic polymers, is widely used in various areas, such as materials engineering and biomedical engineering, due to its superior performance, while 3D printing via fused deposition modeling (FDM) provides a facile method of preparing PE products. To optimize the performance and assess the degradation of FDM-printed PE materials, we systematically investigate the influences of printing parameters, such as fiber diameter (stretching) and printer head temperature, and degradation, such as UV exposure and thermal degradation, on the mechanical performance of FDM-printed PE fibers. When FDM-printed PE fibers with a smaller diameter are prepared under a higher collecting speed, they undergo stronger stretching, and thus, show higher tensile strength and Young’s modulus values. Meanwhile, the tensile strength and Young’s modulus decrease as the printer head temperature increases, due to the lower viscosity, and thus, weaker shearing at high temperatures. However, degradation, such as UV exposure and thermal degradation, cause a decrease in all four mechanical properties, including tensile strength, Young’s modulus, tensile strain and toughness. These results will guide the optimization of FDM-printed PE materials and help to assess the durability of PE products against degradation for their practical application.

## 1. Introduction

Polyethylene (PE), including both high-density polyethylene (HDPE) and low-density polyethylene (LDPE), is an important material polymerized from ethylene and is widely used in science and engineering [1,2,3]. PE has a simple basic structure of (CH2)_n_ and is constructed exclusively from C-C single bonds [4]. PE demonstrates a lot of excellent properties, including low cost, light weight, good flexibility, long durability, optical transparency, electrical insulation, corrosion resistance, good processability and high toughness [5,6]. Due to its superior performance, PE is widely used in daily products [7,8], construction materials [9,10], electronic devices [11], automotive vehicles [12], biomedical devices [13], etc. For example, PE can be made into artificial implants, such as nasal dorsal enhancements, long-term catheters, endoprostheses and facial prosthetics [14,15]. Investigations of the mechanical performance of PE products and their durability against degradation are important, since the product lifetime of PE materials must be taken into account for both safety and budget concerns [16].

The mechanical performance of PE strongly depends on its molecular structure. For example, HDPEs are produced via a low-pressure polymerization process and have an almost linear structure, with a density of 0.941~0.967 g/cm^3^, a crystallinity of 65~85% and a molecular weight of 40,000–300,000 Da, while LDPEs are generally produced via high-pressure polymerization and have many branches, with a density of 0.920~0.923 g/cm^3^, a crystallinity of 45~65% and a molecular weight of 50,000–500,000 Da [17,18]. Compared with LDPEs, HDPEs have a more linear structure, tighter molecular stacking, a higher density, higher crystallinity and high tensile strength, but lower tensile strain [19]. The tensile strength and tensile strain of HDPEs are 28 MPa and 213%, respectively, while those of LDPEs are 10 MPa and 349%, respectively [20].

In addition to molecular structure, the processing parameters of PE products, which could affect the orientational ordering, crystallinity and size of the crystallites in PE products, also strongly influence their mechanical performance. Generally, PE products are processed via injection molding or 3D printing [20,21]. Compared with injection, 3D printing via fused deposition modeling (FDM) provides an opportunity to fabricate customized products on demand and has become more and more popular [22,23,24,25,26,27]. In FDM, PE fibers are extruded out of a tiny nozzle and undergo strong shearing, which is beneficial for orientational ordering and crystallization, and thus, contributes to the high tensile strength and Young’s modulus [28]. Therefore, the printing parameters have a strong influence on the mechanical performance of 3D-printed PE products [29,30].

The excellent mechanical performance of PE could be compromised by degradation [31]. The degradation of PE generally consists of abiotic and biotic degradation [32]. Abiotic degradation caused by light or heat is typically the initial and rate-determining step, leading to smaller molecules [33,34]. UV radiation, for example, leads to the formation of free radicals, which combine with oxygen at the surface and form peroxides and hydroperoxides [35]. So far, there are no systematic studies on the dependence of PE’s mechanical properties on processing parameters and their variations against degradation. Investigations of the mechanical performance of FDM-printed PE products and their durability against degradation are thus essential for the widespread applications of PE.

Here, the mechanical performance of FDM-printed PE products and their durability against degradation are systematically investigated, as they are the fundamental building block of FDM-printed products and could exclude the influences of many factors, such as printing path and bulk structure. HDPE and LDPE fibers, whose molecular structures are shown in Figure 1a, are prepared through 3D printing via fused deposition modeling (FDM), as presented in Figure 1b, and their mechanical performance is optimized in terms of the printing parameters and tested against degradation, as presented in Figure 1c. The influences of molecular structures and printing parameters, such as fiber diameter (stretching) and printer head temperature, on the mechanical performance of FDM-printed fibers are investigated in detail. Since UV and heating are the two most common and important triggering sources for PE degradation, the mechanical performance of HDPE and LDPE fibers after UV exposure and thermal degradation are also systematically studied. The results of this study will shine light on the relationship between molecular structure and mechanical properties, guide the optimization of FDM-printed PE materials and help to assess the durability of PE products against degradation, which will pave the way for their practical application.

## 2. Materials and Methods

### 2.1. Materials

High-density polyethylene (HDPE, 0.941~0.967 g/cm^3^, molecular weight of 40,000–300,000 Da) and low-density polyethylene (LDPE, 0.920~0.923 g/cm^3^, molecular weight of 50,000–500,000 Da) filaments were purchased from Yitailong Hi-tech Material Co., Ltd., Changsha, China.

### 2.2. Sample Preparation

#### 2.2.1. Preparation of FDM-Printed PE Fibers

HDPE and LDPE fibers were prepared via fused deposition modeling (FDM). The polymer feed was melted in the printer head at 215 °C, 235 °C or 255 °C, and was extruded out of the tapered nozzle to form fibers. HDPE and LDPE fibers were collected on a winder at 25 °C and 50% relative humidity, as shown in Appendix A. The polymer feeding rate was kept constant at 0.3 cm/s. HDPE and LDPE fibers with diameters of 200 µm, 400 µm and 600 µm were obtained using collecting speeds of 11.6 cm/s, 3.1 cm/s and 1.2 cm/s, respectively.

#### 2.2.2. Polarized Light Microscopy (PLM)

PLM images were taken using a polarized optical microscope equipped with a digital CCD camera in transmission mode. HDPE and LDPE fibers were placed between a crossed polarizer and analyzer and they were either parallel to the polarizer or positioned at an angle of 40° with respect to the polarizer.

#### 2.2.3. UV Exposure and Thermal Degradation

HDPE and LDPE fibers were cut into 5 cm samples. For UV exposure, HDPE and LDPE samples were placed 8 cm under a UV lamp (395 nm, 18 W) for 12 h, 24 h and 48 h. For thermal degradation, HDPE and LDPE samples were baked in an oven at 100 °C, for 12 h and 24 h. Unless otherwise specified, five samples were tested for each group.

#### 2.2.4. Mechanical Testing

The mechanical performance of HDPE and LDPE fibers was tested using an electronic single yarn strength tester at a speed of 10 mm/min at room temperature. Stress was calculated by dividing the force by the cross-sectional area of the cylindrical fibers. Strain was measured directly from the displacement. Young’s modulus was determined from the slope of stress–strain curves at 1% strain in the linear elastic deformation region. Toughness was calculated by integrating the shadow area of the stress–strain curves. Five samples were tested for each group. Average value and standard deviation were calculated for each group. Statistical significance was analyzed only between the groups of HDPE fibers for clarity.

## 3. Results and Discussion

To prepare the FDM-printed fibers, the HDPE and LDPE fibers were extruded out of a tiny nozzle in the 3D printer and underwent strong shearing during the printing process, which is beneficial for the orientation and crystallization of PE molecules. The birefringence of the HDPE and LDPE fibers was investigated via polarized light microscopy (PLM), as shown in Figure 2. When HDPE and LDPE fibers are placed along the polarizer, they are essentially dark. When HDPE and LDPE fibers make an angle of 40° with respect to the polarizer, both of them show clear birefringence, suggesting the orientational ordering of PE molecules along the fiber direction. Compared with LDPE fibers, HDPE fibers show a more uniform texture in birefringence. However, LDPE fibers appear slightly brighter under PLM, which is attributed to their higher transparency.

### 3.1. Optimization of FDM-Printed Fibers: Influences of Fiber Diameter (Stretching) and Printer Head Temperature

The mechanical performance of FDM-printed fibers strongly depends on the printing parameters, and thus, the influences of fiber diameter (stretching) and printer head temperature were systematically investigated. HDPE and LDPE fibers with different diameters were prepared using different collecting speeds. Their stress–strain curves are shown in Figure 3 and their mechanical properties are analyzed in Figure 4 and Appendix A. Both HDPE and LDPE fibers show similar dependence on the fiber diameter. As the fiber diameter increases from 200 μm to 600 μm, the tensile strength and Young’s modulus of HDPE and LDPE fibers decrease, while their tensile strain increases. This is because PE fibers with a smaller diameter are prepared at a higher collecting speed and undergo more stretching, thus having better molecular ordering and better mechanical performance. Since HDPE and LDPE fibers generally show similar dependence on the printing parameters, statistical significance was analyzed only between the groups of HDPE fibers for clarity. The results demonstrate that 200 μm HDPE fibers show a significant difference between 400 μm and 600 μm HDPE fibers in tensile strength and tensile strain, suggesting that fiber diameter (stretching) is an important printing parameter. The fracture surface morphologies of the 3D-printed HDPE and LDPE fibers with a diameter of 200 μm after tensile fracture are shown in Appendix A. Compared to LDPE fibers, the fracture surface morphology of HDPE fibers is more flat, which is attributed to their higher crystallinity.

The dependence of HDPE and LDPE fibers on their molecular structures could be investigated by comparing their mechanical performance. Since HDPEs have a more linear structure, which facilitates the ordering and crystallization of PE molecules, HDPE fibers overall show higher tensile strength and Young’s modulus values than LDPE fibers. However, the tensile strain of HDPE fibers is smaller than that of LDPE fibers. The toughness of HDPE fibers, which was calculated by integrating the shadow area of the stress–strain curves, is comparable to that of LDPE fibers when the diameters are 200 and 400 μm. This is because HDPE fibers have higher tensile strength but lower tensile strain. But the toughness of LDPE fibers with a diameter of 600 μm is much lower than that of HDPE fibers due to its very low tensile strength.

The influences of printer head temperature on mechanical performance were also investigated by printing 200 μm HDPE and LDPE fibers at 215 °C, 235 °C and 255 °C. Their stress–strain curves are shown in Figure 5 and their mechanical properties are analyzed in Figure 6 and Appendix A. Both HDPE and LDPE fibers show similar dependence on the printer head temperature. Interestingly, their tensile strength and Young’s modulus values decrease as the printer head temperature increases, while their tensile strain increases. A possible reason for these results is that PE that is melted at a higher temperature has lower viscosity, and thus, undergoes weaker shearing, resulting in a less ordered molecular arrangement along the fiber direction.

### 3.2. Degradation of FDM-Printed Fibers: Influences of UV Exposure and Thermal Degradation

In general, degradation will greatly compromise the mechanical performance of materials, while UV exposure is one of the main sources causing the degradation. The influences of UV exposure on the mechanical performance of FDM-printed fibers were systematically investigated by exposing PE fibers to UV for different durations. The stress–strain curves of the HDPE and LDPE fibers are shown in Figure 7 and their tensile strength, Young’s modulus, tensile strain and toughness are analyzed in Figure 8 and Appendix A. As expected, UV exposure will break the PE molecular chains. Therefore, all four mechanical properties of HDPE and LDPE fibers, including tensile strength, Young’s modulus, tensile strain and toughness, decrease as the UV exposure time increases. In particular, after 24 h UV exposure, the tensile strength and Young’s modulus show a significant decrease. Compared to HDPE fibers, LDPE fibers show a larger decrease in their mechanical performance. This is because the crystallinity of HDPE fibers is higher than that of LDPE fibers, and thus, HDPE fibers are more resistant to UV degradation.

In addition to UV exposure, the influences of thermal degradation were also investigated. The stress–strain curves of HDPE and LDPE fibers before and after thermal degradation are shown in Figure 9 and their mechanical properties are summarized in Figure 10 and Appendix A. To avoid the influence of fiber melting and thermal relaxation on mechanical performance, the HDPE and LDPE fibers were baked at 100 °C, which is below the melting temperature of PE, and no observable changes in suspended fibers after the thermal degradation tests ensued. Results similar to those caused by UV exposure are observed for thermal degradation, in that the tensile strength, Young’s modulus, tensile strain and toughness of HDPE and LDPE fibers all decrease as the baking time increases. However, compared with UV exposure, statistical significance starts to appear only after 24 h of baking at 100 °C, suggesting that PE fibers are more susceptible to UV exposure than thermal degradation. In addition, the decrease in the mechanical performance could be attributed to thermal degradation. This is because thermal relaxation generally results in an increase in tensile strain instead of a decrease.

## 4. Conclusions

HDPE fibers overall have higher tensile strength and Young’s modulus values but lower tensile strain than LDPE fibers. The mechanical performance of FDM-printed fibers strongly depend on the printing parameters, such as fiber diameter (stretching) and printer head temperature. Interestingly, the influences of printing parameters are very different from those of degradation. For example, when a smaller fiber diameter and lower printer head temperature cause an increase in the tensile strength and Young’s modulus, they will also cause a decrease in tensile strain, and thus, roughly no change in toughness. However, degradation, such as UV exposure and thermal degradation, generally causes a decrease in all four mechanical properties, including tensile strength, Young’s modulus, tensile strain and toughness. In addition, PE fibers are more susceptible to UV exposure than thermal degradation. These results will shine light on the relationship between molecular structure and mechanical properties, guide the optimization of FDM-printed PE materials and help to assess the durability of PE products against degradation during their practical application.

## Figures and Tables

**Figure 1 materials-16-05182-f001:**
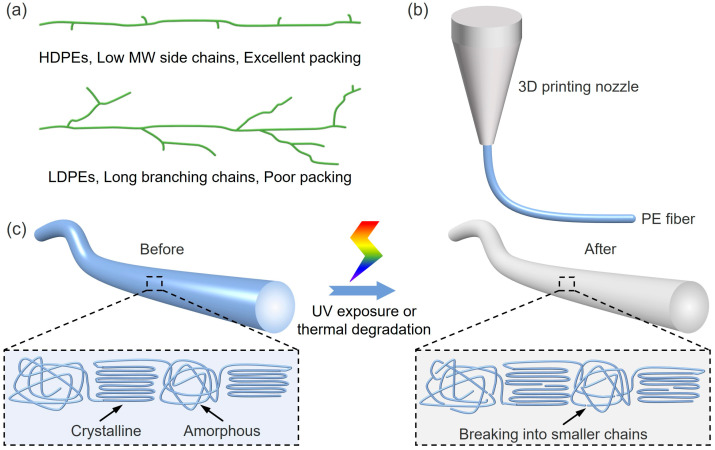
Preparation and degradation of PE fibers. (**a**) Schematics showing the molecular structures of HDPEs and LDPEs. (**b**) Preparation of PE fibers through 3D printing via fused deposition modeling (FDM). (**c**) Schematics showing the microstructures of FDM-printed PE fibers before and after UV exposure or thermal degradation.

**Figure 2 materials-16-05182-f002:**
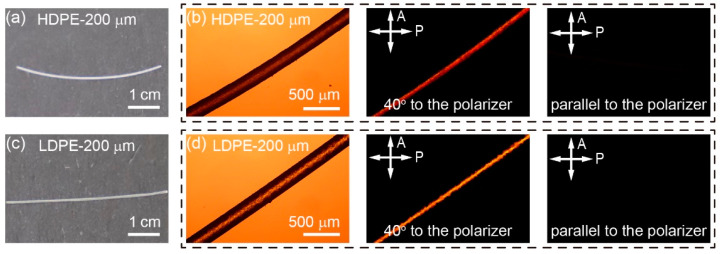
Photographs and PLM images of HDPE and LDPE fibers. (**a**) Photograph of an FDM-printed 200 μm HDPE fiber. (**b**) PLM images of an HDPE fiber parallel to the polarizer or at a 40° angle with respect to the polarizer. (**c**) Photograph of an FDM-printed 200 μm LDPE fiber. (**d**) PLM images of an LDPE fiber parallel to the polarizer or at a 40° angle with respect to the polarizer.

**Figure 3 materials-16-05182-f003:**
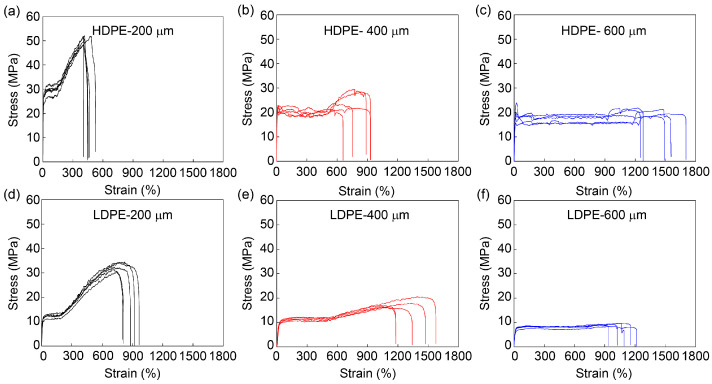
Mechanical performance of HDPE and LDPE fibers with different diameters. Stress–strain curves of HDPE fibers with diameters of (**a**) 200 μm, (**b**) 400 μm and (**c**) 600 μm and LDPE fibers with diameters of (**d**) 200 μm, (**e**) 400 μm and (**f**) 600 μm. The printer head temperature was 215 °C. HDPE and LDPE fibers with a diameter of 200 μm, 400 μm and 600 μm were prepared with collecting speeds of 11.6 cm/s, 3.1 cm/s and 1.2 cm/s, respectively. Five samples were tested for each system, corresponding to five stress–strain curves.

**Figure 4 materials-16-05182-f004:**
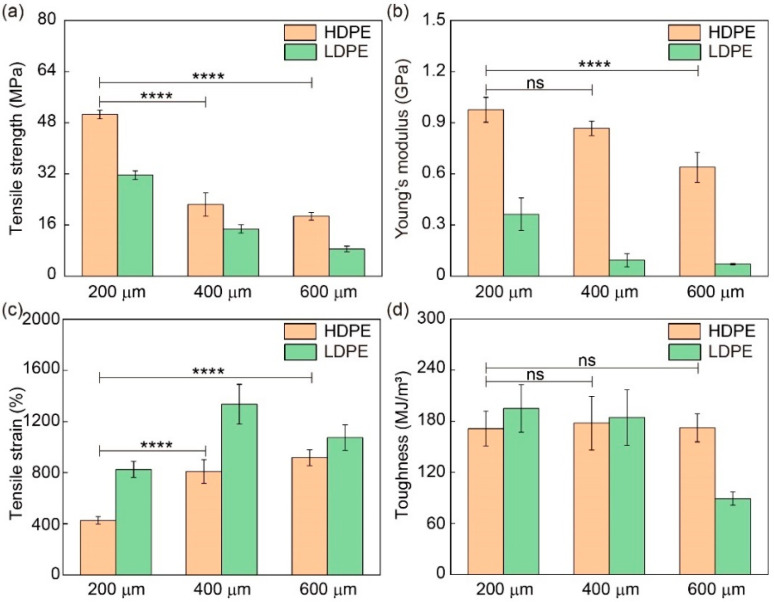
Mechanical properties of HDPE and LDPE fibers with different diameters. (**a**) Tensile strength, (**b**) Young’s modulus (1% strain), (**c**) tensile strain and (**d**) toughness of HDPE and LDPE fibers with diameters of 200 μm, 400 μm and 600 μm. The printer head temperature was 215 °C. HDPE and LDPE fibers with diameters of 200 μm, 400 μm and 600 μm were prepared with collecting speeds of 11.6 cm/s, 3.1 cm/s and 1.2 cm/s, respectively. Unless otherwise specified, Young’s modulus was determined at 1% strain. Unless otherwise specified, statistical significance was analyzed only between the groups of HDPE fibers for clarity. Unless otherwise specified, ns denotes not significant; **** denotes *p* < 0.0001. *p* is the probability that the difference between the two sets of data is caused by experimental error.

**Figure 5 materials-16-05182-f005:**
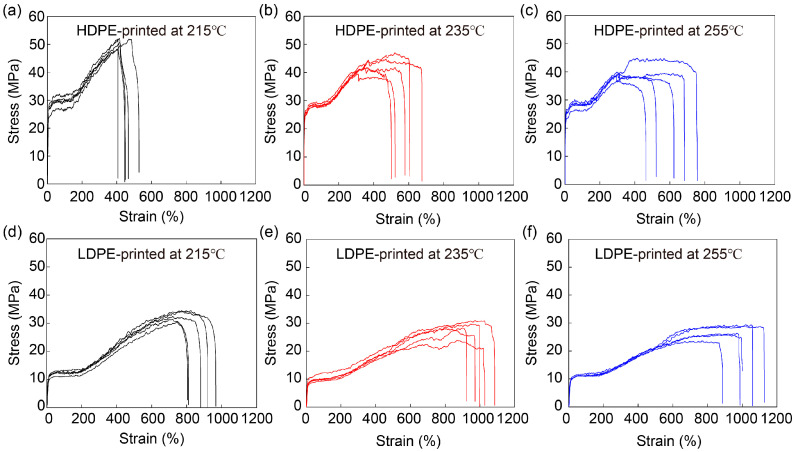
Mechanical performance of HDPE and LDPE fibers printed at different printer head temperatures. Stress–strain curves of HDPE fibers printed at (**a**) 215 °C, (**b**) 235 °C and (**c**) 255 °C and LDPE fibers printed at (**d**) 215 °C, (**e**) 235 °C and (**f**) 255 °C. The diameter of HDPE and LDPE fibers was about 200 μm.

**Figure 6 materials-16-05182-f006:**
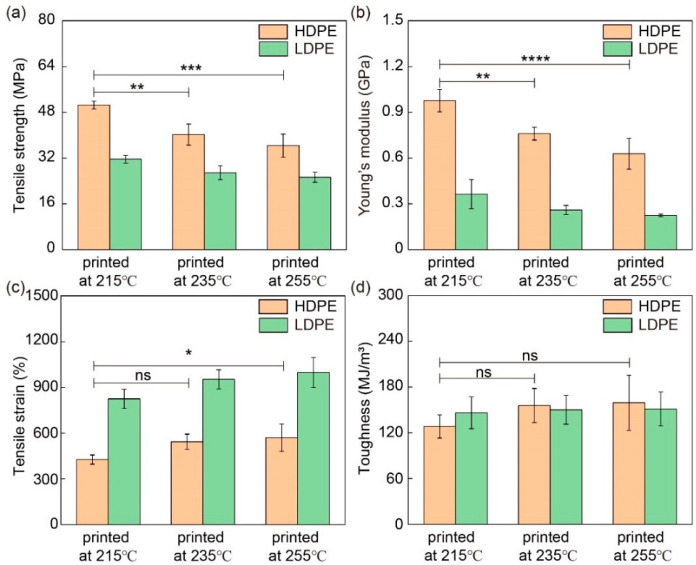
Mechanical properties of HDPE and LDPE fibers printed at different printer head temperatures. (**a**) Tensile strength, (**b**) Young’s modulus (1% strain), (**c**) tensile strain and (**d**) toughness of HDPE and LDPE fibers printed at 215 °C, 235 °C and 255 °C. The diameter of HDPE and LDPE fibers was about 200 μm. ns denotes not significant; * denotes *p* < 0.05; ** denotes *p* < 0.01; *** denotes *p* < 0.001; **** denotes *p* < 0.0001. *p* is the probability that the difference between the two sets of data is caused by experimental error.

**Figure 7 materials-16-05182-f007:**
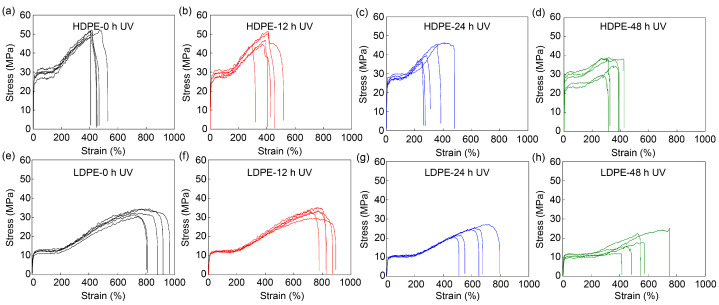
Mechanical performance of HDPE fibers before and after UV exposure. Stress–strain curves of HDPE fibers (**a**) before UV exposure, (**b**) after 12 h, (**c**) after 24 h and (**d**) after 48 h UV exposure, and of LDPE fibers (**e**) before UV exposure, (**f**) after 12 h, (**g**) after 24 h and (**h**) after 48 h UV exposure. The printer head temperature was 215 °C. The diameter of HDPE and LDPE fibers was about 200 μm. The samples were placed 8 cm under a UV lamp (395 nm, 18 W).

**Figure 8 materials-16-05182-f008:**
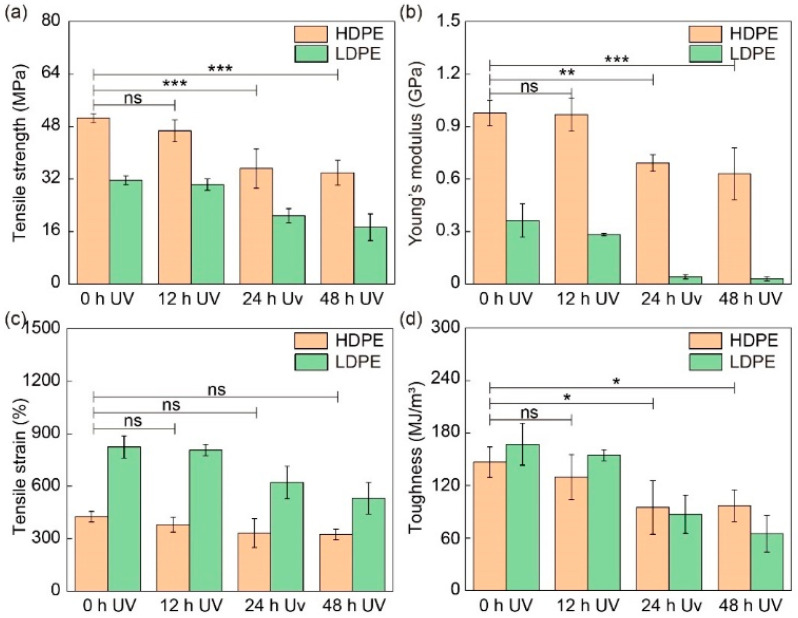
Mechanical properties of HDPE and LDPE fibers before and after UV exposure. (**a**) Tensile strength, (**b**) Young’s modulus, (**c**) tensile strain and (**d**) toughness of HDPE and LDPE fibers before UV exposure and after 12 h, 24 h and 48 h UV exposure. The printer head temperature was 215 °C. The diameter of HDPE and LDPE fibers was about 200 μm. The samples were placed 8 cm under a UV lamp (395 nm, 18 W). ns denotes not significant; * denotes *p* < 0.05; ** denotes *p* < 0.01; *** denotes *p* < 0.001. *p* is the probability that the difference between the two sets of data is caused by experimental error.

**Figure 9 materials-16-05182-f009:**
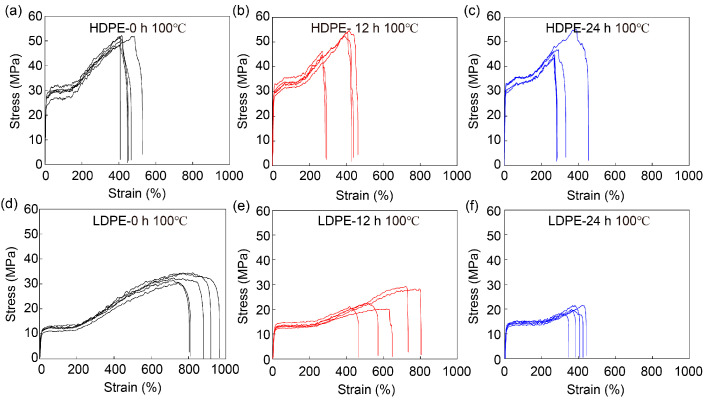
Mechanical performance of HDPE and LDPE fibers before and after thermal degradation. Stress–strain curves of HDPE fibers (**a**) before baking, (**b**) after 12 h and (**c**) after 24 h baking at 100 °C, and of LDPE fibers (**d**) before baking, (**e**) after 12 h and (**f**) after 24 h baking at 100 °C. The printer head temperature was 215 °C. The diameter of HDPE and LDPE fibers was about 200 μm.

**Figure 10 materials-16-05182-f010:**
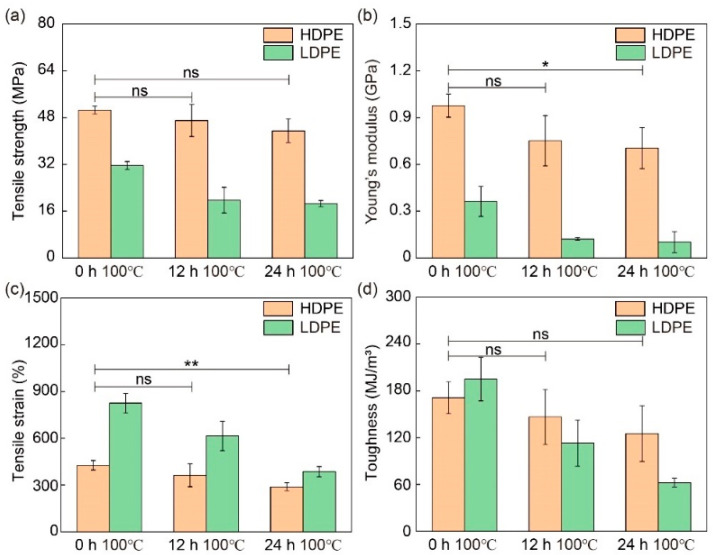
Mechanical properties of HDPE and LDPE fibers before and after thermal degradation. (**a**) Tensile strength, (**b**) Young’s modulus, (**c**) tensile strain and (**d**) toughness of HDPE and LDPE fibers before baking and after 12 h and 24 h baking at 100 °C. The printer head temperature was 215 °C. The diameter of HDPE and LDPE fibers was about 200 µm. ns denotes not significant; * denotes *p* < 0.05; ** denotes *p* < 0.01. *p* is the probability that the difference between the two sets of data is caused by experimental error.

## Data Availability

Data available on request due to restrictions e.g., privacy or ethical.

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
