# Peer review of "Mechanical Performance of 3D-Printed Polyethylene Fibers and Their Durability against Degradation"

_materials, 2023, doi:10.3390/ma16145182_

Round 1
Reviewer 1 Report
While the work is interesting and fits the scope of the journal the analysis of the observed trends is rather disappointing. The results are presented in a very dry form without any explanation neither on the HDPE and LDPE similarities and differences nor on the effect of the tested parameters. Major revisions are required so that the authors should analyze in more detail the findings but also to present better the results.
) As stated above the reported trends are rather poorly analyzed. For example no explanation is given why the LDPE shows such a drastic decrease in the Young´s modulus as a function of exposure time compared to HDPE. Same can be stated for the tensile strain, this time for HDPE. It would be interesting to compare not only the absolute values, which are significantly different between HDPE and LDPE but also on the relative change using as reference for example the 0h time or the 215oC temperature etc. That would make more clear the effect of the variables on the HDPE vs. LDPE.
) In Figure 3 do different curves correspond to different samples? For clarity it would be better to use different color for each sample. Also, for a more direct visual comparison on the effect of diameter but also between HDPE and LDPE the x- and y-scales in all samples could have the same range.
) Do the error bars in Fig. 4 correspond to the standard deviation as obtained from the 5 different samples? Also, it should be denoted the correspondence of the p variable used in the legend.
The term “modeled” used to indicate sketches in specific figure panels (for example lines 81 and 82) could be changed to “are seen” or “are presented”.
Line 117 “if not specified”, do the authors mean “unless otherwise specified” ? The number of samples is repeated with respect to the first instance (line 125, 184).
Line 136: “To prepared” -> “The prepare”.
The manuscript could benefit from some polishing in english syntax, but there is no important problem.
Author Response
While the work is interesting and fits the scope of the journal the analysis of the observed trends is rather disappointing. The results are presented in a very dry form without any explanation neither on the HDPE and LDPE similarities and differences nor on the effect of the tested parameters. Major revisions are required so that the authors should analyze in more detail the findings but also to present better the results.
(1) As stated above the reported trends are rather poorly analyzed. For example no explanation is given why the LDPE shows such a drastic decrease in the Young´s modulus as a function of exposure time compared to HDPE. Same can be stated for the tensile strain, this time for HDPE. It would be interesting to compare not only the absolute values, which are significantly different between HDPE and LDPE but also on the relative change using as reference for example the 0h time or the 215℃ temperature etc. That would make more clear the effect of the variables on the HDPE vs. LDPE.
We thank the reviewer for the helpful comments.PE fibers are composed of crystalline part and amorphous part, as shown in Figure 1(c). The crystalline part mainly provides tensile strength and Young's modulus, while the amorphous part mainly provides tensile strain. Since the crystallinity of HDPE is higher than that of LDPE, the tensile strength and Young's modulus of HDPE decrease less than that of LDPE, but the tensile strain decreases drastically when they are under UV exposure or thermal degradation. And we have added the comparation in our revised manuscript.
(2) In Figure 3 do different curves correspond to different samples? For clarity it would be better to use different color for each sample. Also, for a more direct visual comparison on the effect of diameter but also between HDPE and LDPE the x- and y-scales in all samples could have the same range.
We thank the reviewer for the helpful comments. Different curves correspond to different samples in Figure 3. Different colors are used for each sample for clarity. And we have kept the x- and y-scales in all samples in same ranges in our revised manuscript.
(3) Do the error bars in Fig. 4 correspond to the standard deviation as obtained from the 5 different samples? Also, it should be denoted the correspondence of the p variable used in the legend.
We thank the reviewer for the helpful comments. As the reviewer understand, the error bars in Figure 4 correspond to the standard deviation as obtained from the 5 different samples. P is the probability that the difference between the two sets of data is caused by experimental error. The smaller the value of P is, the smaller the probability of differences between the two sets of data caused by experimental errors is.
(4) The term “modeled” used to indicate sketches in specific figure panels (for example lines 81 and 82) could be changed to “are seen” or “are presented”.
We thank the reviewer for the helpful comments. We have corrected the term “modeled” into “presented” in our revised manuscript.
(5) Line 117 “if not specified”, do the authors mean “unless otherwise specified”? The number of samples is repeated with respect to the first instance (line 125, 184).
We thank the reviewer for the helpful comments. We have deleted the repeated sentence in our revised manuscript.
(6) Line 136: “To prepared” -> “The prepare”.
We thank the reviewer for the helpful comments. We have corrected this typo in our revised manuscript.

Reviewer 2 Report
The authors report a study on the mechanical performances of 3D-printed polyethylene fibers and their durability against degradation. Even if the reported issues could have relevance in material science, I believe that before this manuscript can be considered for publication in this Journal the novelty of the work needs to be better addressed.
Many papers have recently reported the mechanical properties of 3D-printed plastic samples (see justa as example, Materials, 2023, 16(8), 3268 or Polymers, 2023, 15(8), 1846). What's new in the methods used for 3D-printed polyethylene fibers?
Fracture surface morphology of 3D-printed specimens after the tensile strength should be provided.
Supplementary materials should be shifted in a separate file.
Minor editing of English language required
Author Response
The authors report a study on the mechanical performances of 3D-printed polyethylene fibers and their durability against degradation. Even if the reported issues could have relevance in material science, I believe that before this manuscript can be considered for publication in this Journal the novelty of the work needs to be better addressed.
Many papers have recently reported the mechanical properties of 3D-printed plastic samples (see justa as example, Materials, 2023, 16(8), 3268 or Polymers, 2023, 15(8), 1846). What's new in the methods used for 3D-printed polyethylene fibers?
We thank the reviewer for the helpful comments. We have added the two references in our revised manuscript. Compared to other 3D-printed plastic samples, PE fibers are prepared by 3D printing via fused deposition modeling (FDM) like melt spinning. When FDM-printed PE fibers are prepared under a higher collecting speed (Figure S1(a)), they undergo a strong stretching in the tapered nozzle which is absent in other 3D-printed plastic samples and melt spinning samples, and thus show higher tensile strength and Young’s modulus. And the shear force in the tapered nozzle is tunable by adjusting the collecting speed. It is a facile method to prepare mechanical strong PE products.
Fracture surface morphology of 3D-printed specimens after the tensile strength should be provided.
We thank the reviewer for the helpful comments. We have provided the fracture surface morphology of 3D-printed specimens (HDPE-200 μm and LDPE-200 μm) in Figure S2 in our revised manuscript. Compared to LDPE fibers, the fracture surface morphology of HDPE fibers are more flat at-tributed to their higher crystallinity.
Supplementary materials should be shifted in a separate file.
We thank the reviewer for the helpful comments. We have shifted the supplementary materials in a separate file.

Reviewer 3 Report
See the attached pdf.

Please check the text again. In my review I highighted some typical typos of the paper. Besides, usage of commas could be overviewed too.
Author Response
3D printing is becoming more and more popular, which means that its application areas are very rich. People who want to 3D print something can be novices or professionals in this field. There are many people who cannot analyze the filament properties due to special knowledge and devices. In this aspect, the article is interesting and can attract the attention of many people. The structure of the paper is correct, although Section 5. seems to be empty. I have some questions:
(1) If I understood correctly you purchased two types of PE filament for 3D printing. Then you used an FDM printer (what kind of?) to create fibers. What did you do with the hot material leaving the nozzle? How did you use a winder to store it? For a test, you created samples of 5 cm. If you wind up the hot fiber and let it cool down, how you do create the samples? I would welcome more detailed description about producing the fibers and samples. Maybe with some photos.
We thank the reviewer for the helpful comments. As the reviewer understood, we purchased HDPE and LDPE filament for 3D printing via an FDM printer (Raise 3D N1, China) to prepare HDPE and LDPE fibers as shown in Figure S1. The filament is melted in a hot printer head, and extruded out of the tapered nozzle to create PE fibers like melt spinning. Finally, PE fibers are collected on a winder rotated at a constant rotational speed. PE fibers with different diameters are obtained with different rotational speeds.
(2) In line 96: I would use filament instead of fiber. It would be more clear to use two terms to name the material before and after printing.
We thank the reviewer for the helpful comments. We have corrected the “filament” into “fiber” in our revised manuscript.
(3) HDPE requires a temperature between 230 °C and 260 °C to flow consistently. Why did you test it at 215°C? (line 101)
We thank the reviewer for the helpful comments. The HPDE produced by each manufacturer are different, varied in density, molecular weight and crystallinity. The temperature recommended by the manufacturer of the HDPE filaments for 3D printing we purchased is 175-230 ° C, so we test it at 215°C.
(4) LDPE requires a nozzle of about 120°C, as far as I know. Have you tested it above 200°C?
We thank the reviewer for the helpful comments. We have tested LDPE at 215 ℃, 235 ℃ and 255 ℃, and LDPE fibers can be prepared continuously and collected on a winder, as shown in Figure S1(b) where the LDPE fibers are prepared at 215 ℃.
(5) I would recommend a spell check too. Some typos:
- line 60: inject molding - > injection
- line 72, 171: dependance ->dependence
- line 136: To prepared -> To prepare
- line 141: angle of 40o
- line 255,282: PE fibers is more -> PE fibers are more
- line 258: instead of decrease -> instead of a decrease.
We thank the reviewer for the helpful comments. We have corrected the typos in our revised manuscript.

Reviewer 4 Report
Please, see the attached document.

Author Response
Reference: Materials 2447795
Title: Mechanical Performances of 3D-Printed Polyethylene Fibers and Their Durability against
Degradation
Authors: Yao Xiao, Shikai Zhang, Jingyi Chen, Baoling Guo, Dong Chen
Decision: minor revision
Comments
The manuscript aims to investigate comparatively the mechanical behavior of HDPE and LDPE fibers obtained by 3D printing (fused deposition modeling). In this sense the authors followed the tensile strength, tensile strain, Young modulus and toughness of various HDPE/LDPE as a function of the fiber diameter and printer head temperature. Also their temperature and UV stability is monitored. The investigation is not spectacular, however the manuscript is well-written, very neat and the results could be important for regular users of PE.
Comments
In the introduction the authors summarize the main aspects of polyethylene: structure, applications, importance of mechanical performance and influence of the processing procedure on the future characteristics, endorsed with representative references. The aim of the study is clearly stated.
(1) Part 3.1., lines 175-176: “Therefore, the toughness of HDPE fibers is comparable to that of LDPE fibers.” Looking in the Table S1, the toughness values are not quite similar. What was the rationale of the authors for this conclusion?
We thank the reviewer for the helpful comments. We have corrected this conclusion in our revised manuscript. The toughness of HDPE fibers, which is calculated by integrating the shadow area of stress-strain curves, is comparable to that of LDPE fibers when the diameters are 200 and 400 μm. This is because HDPE fibers have a higher tensile strength but a lower tensile strain. But the toughness of LDPE fibers with a diameter of 600 μm is much lower than HDPE fibers for its very low tensile strength.
(2) The Table S2 includes only the values of LDPE, those for HDPE are not included.
We thank the reviewer for the helpful comments. We have added the values of HDPE to Table S2 in our revised manuscript.
(3) Part 3.2., lines 256-257: “In addition, the decrease in mechanical performances could be attributed to thermal degradation other than thermal relaxation.” Thermal relaxation is not at all degradation. This phrasing needs to be reconsidered.
We thank the reviewer for the helpful comments. As the reviewer suggested, thermal relaxation is not at all degradation. And we have corrected the phrasing in our revised manuscript.

Round 2
Reviewer 1 Report
While authors have addressed my comments, in some instances the corresponding information is not available in the manuscript. For example, in Figure 3 (or the corresponding discussion) it should appear a statement in the lines of “for a given system different curves correspond to the five different samples”.
Then, in lines 196-197, “denotes p<0.05”, the definition of p is absent. Even when a definition is trivial it should appear once mentioned to increase the clarity of the manuscript.
It is not clear what is meant by “does not show a good pattern” (line 177)
(line 234): “. And … exposure” -> “and they both break into smaller chains under UV exposure.”.
(line 235): “than that” -> “than these”.
Some erros in the language are indicated in the report. A careful final edit could increase further the clarity of the manuscript.
Author Response
While authors have addressed my comments, in some instances the corresponding information is not available in the manuscript. For example, in Figure 3 (or the corresponding discussion) it should appear a statement in the lines of “for a given system different curves correspond to the five different samples”.
We thank the reviewer for the helpful comments. We have added the statement in our revised manuscript.
Then, in lines 196-197, “denotes p<0.05”, the definition of p is absent. Even when a definition is trivial it should appear once mentioned to increase the clarity of the manuscript.
We thank the reviewer for the helpful comments. We have added the definition of p in our revised manuscript.
It is not clear what is meant by “does not show a good pattern” (line 177)
We thank the reviewer for the helpful comments. The toughness of HDPE fibers, which is calculated by integrating the shadow area of stress-strain curves, is comparable to that of LDPE fibers when the diameters are 200 and 400 μm. This is because HDPE fibers have a higher tensile strength but a lower tensile strain. But the toughness of LDPE fibers with a diameter of 600 μm is much lower than HDPE fibers for its very low tensile strength. And we have corrected conclusion in our revised manuscript.
(line 234): “. And … exposure” -> “and they both break into smaller chains under UV exposure.”.
We thank the reviewer for the helpful comments. We have corrected this typo in our revised manuscript.
(line 235): “than that” -> “than these”.
We thank the reviewer for the helpful comments. We have corrected this typo in our revised manuscript.